

# Whole-transcriptome analysis reveals mechanisms underlying antibacterial activity and biofilm inhibition by a malic acid combination (MAC) in *Pseudomonas aeruginosa*

Kunping Song[1,*], Li Chen[2,*], Nanhua Suo[1], Xinyi Kong[1], Juexi Li[1], Tianyu Wang[1], Lanni Song[3], Mengwei Cheng[3], Xindian Guo[4], Zhenghe Huang[4], Zichen Huang[4], Yixin Yang[1,3,5], Xuechen Tian[3,5] and Siew Woh Choo[1,3,5]

[1] Wenzhou-Kean University, College of Science, Mathematics and Technology, Wenzhou, Zhejiang, China
[2] Universiti Malaya, Institute of Biological Sciences, Faculty of Science, Kuala Lumpur, Kuala Lumpur, Malaysia
[3] Wenzhou-Kean University, Wenzhou Municipal Key Laboratory for Applied Biomedical and Biopharmaceutical Informatics, Wenzhou, Zhejiang, China
[4] Wenzhou No.2 Foreign Language School, Wenzhou, Zhejiang, China
[5] Wenzhou-Kean University, Zhejiang Bioinformatics International Science and Technology Cooperation Center, Wenzhou, Zhejiang, China
* These authors contributed equally to this work.

Corresponding authors
Xuechen Tian,
tianxuechen@wku.edu.cn
Siew Woh Choo, cwoh@wku.edu.cn

## ABSTRACT

**Background**. *Pseudomonas aeruginosa* is a highly prevalent bacterial species known for its ability to cause various infections and its remarkable adaptability and biofilm-forming capabilities. In earlier work, we conducted research involving the screening of 33 metabolites obtained from a commercial source against two prevalent bacterial strains, *Escherichia coli* and *Staphylococcus aureus*. Through screening assays, we discovered a novel malic acid combination (MAC) consisting of malic acid, citric acid, glycine, and hippuric acid, which displayed significant inhibitory effects. However, the precise underlying mechanism and the potential impact of the MAC on bacterial biofilm formation remain unknown and warrant further investigation.

**Methods**. To determine the antibacterial effectiveness of the MAC against *Pseudomonas aeruginosa*, we conducted minimum inhibitory concentration (MIC) and minimum bactericidal concentration (MBC) assays. Transmission electron microscopy (TEM) and scanning electron microscopy (SEM) techniques were employed to observe bacterial morphology and biofilm formation. We further performed a biofilm inhibition assay to assess the effect of the MAC on biofilm formation. Whole-transcriptome sequencing and bioinformatics analysis were employed to elucidate the antibacterial mechanism of the MAC. Additionally, the expression levels of differentially expressed genes were validated using the real-time PCR approach.

**Results**. Our findings demonstrated the antibacterial activity of the MAC against *P. aeruginosa*. SEM analysis revealed that the MAC can induce morphological changes in bacterial cells. The biofilm assay showed that the MAC could reduce biofilm formation. Whole-transcriptome analysis revealed 1093 differentially expressed genes consisting of 659 upregulated genes and 434 downregulated genes, in response to

the MAC treatment. Mechanistically, the MAC inhibited *P. aeruginosa* growth by targeting metabolic processes, secretion system, signal transduction, and cell membrane functions, thereby potentially compromising the survival of this human pathogen. This study provides valuable insights into the antibacterial and antibiofilm activities of the MAC, a synergistic and cost-effective malic acid combination, which holds promise as a potential therapeutic drug cocktail for treating human infectious diseases in the future.

# INTRODUCTION

*Pseudomonas aeruginosa*, a Gram-negative bacterium, is frequently found in soil and water habitats (*Wu et al., 2015*). This bacterium is recognized as an opportunistic pathogen and is among the most prevalent hospital-acquired infections worldwide, particularly in individuals with compromised immune systems (*Kerr & Snelling, 2009*). *P. aeruginosa* is a ubiquitous culprit in various infections including, burn-wound infections (*Norbury et al., 2016*), urinary tract infections (*Narten et al., 2012*), surgical site infection (*Elgohari et al., 2017*), ventilator-associated pneumonia (*Kalil et al., 2016*), organ transplants infections (*Ruiz et al., 2006*), and in severe cases, bloodstream infections and sepsis (*Micek et al., 2011*; *Thaden et al., 2017*). Furthermore, *P. aeruginosa* can adapt and survive in different environments and can adhere to various surfaces to form biofilms, including food industry equipment (*Coughlan et al., 2016*) and medical materials (*Ghafoor, Hay & Rehm, 2011*). Forming biofilm is regarded as a crucial factor determining the pathogenicity of *P. aeruginosa* infection (*Wareham & Curtis, 2007*). As a complex community of cells growing on a surface, bacterial cells residing within biofilms have the ability to evade and circumvent host immune responses (*Lewis, 2001*) and exhibit up to a thousand-fold more resistance to antimicrobial agents than their planktonic counterparts (*Kerr & Snelling, 2009*). Given the widespread occurrence of *P. aeruginosa* and the limited efficacy of current treatments, compounded by the emergence and dissemination of multidrug-resistant and extensively drug-resistant strains (*Sader et al., 2018*), the World Health Organization has acknowledged this pathogen as crucial research focus on developing alternative therapeutic interventions and novel treatment approaches (*Tacconelli et al., 2018*).

Living organisms are valuable sources of antibacterial compounds (*Álvarez-Martínez, Barrajón-Catalán & Micol, 2020*). Among the natural sources, organic acids are found in animals, plants, and microorganisms and have received increased attention for their biological properties and potential applications as antibacterial agents. For instance, malic acid, citric acid, and hippuric acid have been reported to possess antibacterial properties when used alone (*Amrutha, Sundar & Shetty, 2017*; *Borah et al., 2023*). Moreover, combinations of organic acids or other substances can enhance the antibacterial properties of organic acids and reduce the likelihood of resistance development by targeting multiple

bacterial functions (*Adamczak, Ozarowski & Karpinski, 2020*; *Feng et al., 2010*; *Kovanda et al., 2019*).

In the present study, we embarked on a research endeavor aimed at identifying antibacterial compounds. Our investigation involved the screening of 33 metabolites, which were procured from a commercial source. The screening assays were conducted to evaluate the efficacy of these metabolites against two prevalent bacterial strains (*Escherichia coli* and *Staphylococcus aureus*). Remarkably, our screening assays unveiled a novel malic acid combination (MAC), consisting of malic acid, citric acid, glycine, and hippuric acid, which exhibited prominent inhibitory effects (Tian et al., 2023, unpublished data). The significance of this discovery is further substantiated by the China Invention Patent Number CN202111195294.4 and the Luxembourg Invention Patent Number LU102887. Intriguingly, our data demonstrated that the antibacterial effect of the MAC surpassed that of its individual constituent metabolites, implying the possibility of a synergistic effect of these organic acids against a wide spectrum of bacteria. However, despite the remarkable findings, the antibacterial effect of MAC against *P. aeruginosa* and the precise underlying mechanism remains unknown. Furthermore, the potential impact of the MAC on bacterial biofilm formation has yet to be investigated.

Here we examined the antibacterial activity, antibiofilm activity and underlying mechanism of the MAC. The antibacterial efficacy of the MAC was assessed through antibacterial assays and biofilm inhibition assays, followed by morphological analysis of *P. aeruginosa*. RNA-Seq approach was employed to elucidate the transcriptomic response of *P. aeruginosa* to the MAC treatment. Our results revealed that the MAC can inhibit the growth of various bacterial cells and impede the formation of biofilms in *P. aeruginosa*. RNA-Seq data revealed that the MAC suppresses the bacterial metabolic processes, secretion system, signal transduction, and cell membrane functions. By harnessing the innate antimicrobial mechanisms of the MAC and leveraging natural resources, this study presents a sustainable and efficacious means to combat bacterial infections in humans.

## MATERIALS AND METHODS

### Malic acid combination (MAC) formulation

All organic acids were procured from Psaitong Beijing, China, in a solid form. The initial concentration of the MAC was used and designated as "C". The composition of the MAC mixture at concentration C is 12 mg/mL malic acid, 2 mg/mL citric acid, 1 mg/mL glycine, and 3 mg/mL hippuric acid. The stock solution was prepared by the two-fold dilution method in eight different concentrations of 4C, 2C, C, 0.5C, 0.25C, 0.125C, 0.0625C, and 0.03125C.

### Bacterial strain, growth media, and culture conditions

The antibacterial efficacy of the MAC was tested using four types of bacteria strains, including *Escherichia coli* ATCC 25922, *Staphylococcus aureus* ATCC 25923, *Pseudomonas aeruginosa* ATCC 27853, and *Serratia marcescens* BNCC 107931. The bacterial strains utilized in this study were sourced from Dr. Aloysius Wong and Dr. Bo Zhang at Wenzhou-Kean University, China. These strains were initially procured from the public
culture collection center. All bacterial cell lines were cultured and propagated in two mL Luria-Bertani broth (LB; Qingdao Hope Bio-Technology Co., Ltd., Qingdao, China) and incubated at 37 °C with 200 rpm.

## Assessment of the minimum inhibitory concentration (MIC) and minimum bactericidal concentration (MBC)

The minimum inhibitory concentration (MIC) of the MAC was assessed using the sterile 96-well polystyrene microtiter plate according to the previously described method (*Wiegand, Hilpert & Hancock, 2008*). The logarithmic growth phase of the bacterial broth was adjusted to a concentration of 0.5 McFarland turbidity standard ($1 \times 10^8$–$2 \times 10^8$ CFU/mL). Approximately $5 \times 10^4$ CFU in 3 µL bacterial suspension was added to 180 µL LB broth medium. Subsequently, 20 µL serial two-fold dilutions of the MAC in concentrations in the order of 4C to 0.03125C were added to columns 3–10. A volume of 200 µL of LB broth medium was introduced into column 1 to serve as the negative control. The positive control, designated as column 2, solely consisted of LB broth subsequently introduced with bacteria. The optical density (OD) was determined at 600 nm using a Varioskan Flash microplate reader (Thermo Fisher Scientific) after 3 h, 6 h, 9 h, 12 h, and 24 h of incubation at 37 °C with 200 rpm. The OD for each replicate well at negative control was subtracted from the OD of the same replicate well at different times. The MIC endpoint is determined as the minimum concentration of the MAC at which there is no observable growth in the test tubes. The growth inhibition at each MAC dilution was measured using the formula: % growth = (OD test well−OD negative control well/OD of corresponding positive control well−OD negative control well/)×100%. In addition, the BIOMYC-3 antibiotic solution (antibiotic ciprofloxacin) (Biological Industries, Kibbutz Beit-Haemek, Israel) was used as a positive control at a concentration of 0.01 mg/mL, and the same volume as MAC under its MIC was incubated with *P. aeruginosa* at 37 °C and 200 rpm for 24 h.The OD value was measured at 600 nm using a Varioskan Flash microplate reader (Thermo Fisher Scientific, Waltham, MA, USA).

The minimum bactericidal concentration (MBC) refers to the lowest concentration of antibacterial agent that kills the test bacteria. After determination of the MIC of the MAC, 50 µL aliquots of all tubes with no visible bacterial growth were inoculated onto LB agar plates (Qingdao Hope Bio-Technology Co., Ltd., Qingdao, China) and incubated at 37 °C for 24 h. The MBC endpoint was identified as the lowest concentration of the MAC at which no visible colonies were observed. All experiments were repeated in triplicate.

## Assessing bacterial ultrastructural alterations through TEM

To explore the bactericidal mechanism of the MAC, TEM was employed to assess ultrastructural damage and membrane integrity in *P. aeruginosa* (*Bouhdid et al., 2010*). The bacterial species were subjected to the MIC (0.2C) concentration, and four groups were formed including two treatment groups and two control groups. Initially, bacterial suspensions were treated with 2.5% glutaraldehyde and fixed for 24 h at 4 °C. The fixed cells were harvested by centrifugation at 4,000× g for five minutes and washed with 0.1M PBS (pH = 7.2) thrice. Gradually increasing concentrations of ethanol ranging from
30%, 50%, 70%, 80%, 90% to 95% were used for dehydration of the samples for 15 min, followed by 100% ethanol for 20 min. All samples were fixed with pure acetone for 20 min. Samples were subjected to a mixture of embedding medium and acetone (V/V = 1/1) for 1 h; samples were then treated with a mixture of embedding medium and acetone (V/V = 3/1) for 3 h; samples were treated overnight with the pure embedding medium. The infiltration-treated samples were embedded and heated overnight at a temperature of 70 °C to acquire the embedded samples. The samples were sectioned using an ultramicrotome (HT7800; Leica, Ltd., Wetzlar, German) to obtain sections ranging from 70–90 nm, which were then stained with lead citrate solution and 2% uranyl acetate solutions for a period of 10 min each. The samples were dried and examined using a transmission electron microscope (HT7800; Hitachi, Ltd., Tokyo, Japan). The images were captured using an accelerating voltage of 10.0 kV under high vacuum conditions.

**Biofilm inhibition assay and SEM analysis**

The biofilm inhibition assay of the MAC was conducted in the sterile 96-well polystyrene microtiter plate as described previously (*Macia, Rojo-Molinero & Oliver, 2014*). Different concentrations (4C to 0.125C) of the MAC were loaded into wells containing 180 µL of Tryptic Soy Broth (TSB; Qingdao Hope Bio-Technology Co., Ltd, Qingdao, China). Then, 3 µL of bacterial suspension containing approximately $5 \times 10^4$ CFU within the logarithmic growth phase was added to each well, and the plate was incubated at 37 °C for 24 h. Distilled water was used as a control for cells treated. After 24 h, the plate was inverted, and the contents of each well were decanted without disturbing the biofilm. The excess planktonic cells were removed using 0.1 M phosphate-buffered saline (PBS, pH = 7.3). The biofilm adhering to each well was fixed with 99% methanol for 20 min, washed once with PBS, and stained with 1% aqueous crystal violet (CV) solution for 20 min. The unbound crystal violet stain was then gently rinsed with PBS. The plate was air-dried at room temperature for 1 h, and OD was measured at 570 nm using a Varioskan Flash microplate reader (Thermo Scientific, China). The experiments were performed in 3 biological replicates, each with 4 technical replicates.

To evaluate the biofilm biomass and observe alterations in biofilm structure following treatment with the MAC, SEM analysis was performed on *P. aeruginosa*. Briefly, *P. aeruginosa* cultures were incubated untreated for 6 h to allow biofilm formation on the tube wall. The cells were then treated with the MIC (0.2C) of the MAC and subsequently incubated at 37 °C for 24 h, in comparison to control samples. The content was decanted slightly to prevent damage to the biofilm, and bacterial cells were fixed with 2.5% glutaraldehyde and dehydrated by filtration in an ethanol solution. The samples were then fixed with a 1% osmic acid solution for 2 h, and the osmic acid waste solution was carefully removed. Samples were rinsed three times, for 15 min each time, with 0.1M PBS (pH = 7.3). The samples were dehydrated in a gradient of 30%, 50%, 70%, 80%, 90%, and 95% ethanol for 15 min each, followed by two 20-minute washes in 100% ethanol. The biofilms were then critical point dried (K850; Quorum Technologies Ltd., Lewes, UK) and sputter-coated with gold using an ion sputtering apparatus (MC1000; Hitachi, Ltd., Tokyo, Japan). SEM micrographs were acquired using a Field emission scanning electron
microscope (SU8010; Hitachi, Ltd., Tokyo, Japan). The images were obtained by subjecting them to high vacuum conditions with an accelerating voltage of 3.0 kV.

## RNA extraction, library construction and sequencing

Analysis was conducted using three MAC-treated *P. aeruginosa* cells samples (treatment group) and three untreated *P. aeruginosa* cell samples as controls. *P. aeruginosa* was exposed to using sub-MIC (0.1 C) of the MAC for 24 h. Total RNA from each sample was performed using the Trizol Reagent (Qiagen, Hilden, Germany). Each sample group has three biological replicates. The concentration and purity of the obtained total RNA were determined using the Nanodrop 2000 (Thermo Fisher Scientific Inc., Waltham, MA, USA), while the RNA integrity was evaluated through the use of the Agilent 2200 Bioanalyzer (Agilent Technologies, USA).

RNA with high quality was used for library construction. The QIAseq FastSelect − 5S/16S/23S Kit (Qiagen, Valencia, CA, USA) was employed to eliminate ribosomal RNA from the total RNA. The resulting RNA was fragmented and reverse-transcribed to generate the first strand of cDNA. The synthesis of the first strand cDNA involved the use of random primers and Actinomycin D. Subsequently, the double-stranded cDNA was purified and subjected to end repair and dA-tailing in a single reaction, followed by T-A ligation to add adaptors to both ends. Size selection of the adaptor-ligated DNA was performed using beads to isolate fragments of approximately 400 bp. Polymerase Chain Reaction (PCR) amplification was carried out for each sample using P5 and P7 primers, followed by bead-based clean-up. Validation was performed using a Qsep100 (Bioptic, Taiwan, China), and quantification was conducted using a Qubit3.0 Fluorometer (Invitrogen, Carlsbad, CA, USA). All libraries were sequenced on the Illumina Novaseq 6000 platform using a $2 \times 150$ paired-end strategy.

## Data processing, differential gene expression and functional enrichment analysis

All raw data generated by sequencing were filtered by Cutadapt (v 1.9.1) (*Martin, 2011*) to remove contamination and low-quality data such as adapter sequence and quality of base less than 20%. To examine the quality of our sequencing data, all clean readswere remapped onto the reference genome of *P. aeruginosa* ATCC 27853 (NCBI reference sequence, GCF_001687285.1) by using Bowtie2 (version: 2.2.6) using default parameters (*Langmead & Salzberg, 2012*). Transcripts in FASTA format were derived from a pre-existing GFF annotation file and appropriately indexed. The indexed file was used as a reference gene file for HTSeq (version: 0.6.1) (*Anders, Pyl & Huber, 2015*) to estimate gene expression levels from the clean pair-end data. Differential expression analysis was performed using the DESeq2 Bioconductor package (*Love, Huber & Anders, 2014*), which compared gene expression levels between the control group and the treatment group to identify differentially expressed genes (DEGs). Genes with |log2 (Fold Change)| ≥ 1 and Padj <0.05 were considered as DEGs. DEGs with a log2(fold change) ≥ 1 were considered upregulated genes, whereas DEGs with a log2(fold change) ≤ −1 were considered downregulated genes. The Gene Ontology (GO) enrichment analysis was

conducted using GOSeq software (version: 1.34.1) (*Young et al., 2010*) with Gene Ontology Resource (*Ashburner et al., 2000*). GO terms with Padj < 0.05 were considered significant. Pathway enrichment analysis was performed using KOBAS software (version: 2.0) (*Xie et al., 2011*), treating Kyoto Encyclopedia of Genes and Genomes (KEGG) (*Kanehisa & Goto, 2000*) pathways as individual units. Pathway with Padj < 0.05 was considered significant.

### Real-time polymerase chain reaction (RT-PCR)

To validate the list of DEGs, RT-PCR analysis was performed on a random set of DEGs and functionally relevant genes. The primer pairs used for this analysis are listed in Table S1. In order to normalize the mRNA levels, the *rpoD* housekeeping gene was used as the control (*Savli et al., 2003*). Total RNA extraction was performed using Trizol reagent, followed by reverse transcription into cDNA using the PrimeScript RT Master Mix reagent kit from Takara (Japan). RT-PCR was performed using the SYBR Green Master Mix (Yeasen Bio-Technology Co., Ltd, Shanghai, China) and the Applied Biosystems QuantStudio 6 Flex Real-Time PCR System (Applied Biosystems, Waltham, MA, USA).

### Statistical analysis

Statistical results of the antibacterial rate and their growth curves were quantitatively expressed as mean $\pm$ SEM (Standard Error of the Mean) based on three independent biological replicates. The differences between the groups' treatment and control were operated by unpaired $t$-test and were considered statistically significant at $P < 0.05$. Statistical analysis of the antibacterial rate, the growth curve, and biofilm inhibition of *P. aeruginosa* were performed using GraphPad Prism 9.0 (GraphPad Software Inc., La Jolla, CA, USA).

## RESULTS

### The MAC exhibits antibacterial activity and alters the morphology of bacteria

To evaluate the efficacy of the MAC against four prevalent pathogenic bacteria (*E. coli*, *S. aureus*, *P. aeruginosa*, and *S. marcescens*), minimum inhibitory concentration (MIC) and minimum bactericidal concentration (MBC) were performed. The findings of the investigation revealed that all the tested bacterial strains exhibited comparable MIC and MBC values (Table 1), which were determined to be 0.2C–0.4C (the variable denoted by "C" was established by the initial concentration of the MAC). The results demonstrated the antimicrobial efficacy of the MAC, which was found to inhibit the growth of all four bacterial species tested at various incubation periods (Figs. 1A–1D). Notably, a reduction in bacterial growth was observed after 24 h of exposure to the MAC (Figs. 1E–1H). In addition, in order to investigate the difference in the effects of MAC and common antibiotics on *P. aeruginosa*, 0.01 mg/mL of ciprofloxacin was used as a control. After 24 h of incubation, the antibacterial effect of 0.2C MAC and ciprofloxacin was found to be comparable significantly inhibiting bacterial growth (Fig. S1). The ultrastructural changes of *P. aeruginosa* cells were evaluated using TEM. The untreated cells displayed well-formed, clearly defined cell walls and membranes, as well as uniformly dense cytoplasm

**Table 1  Antibacterial activity of the MAC.**

| Bacterial strain | MIC | MBC |
| --- | --- | --- |
| *E. coli* | 0.2C | 0.2C |
| *S. aureus* | 0.2C | 0.2C |
| *P. aeruginosa* | 0.2C | 0.2C |
| *S. marcescens* | 0.4C | 0.4C |

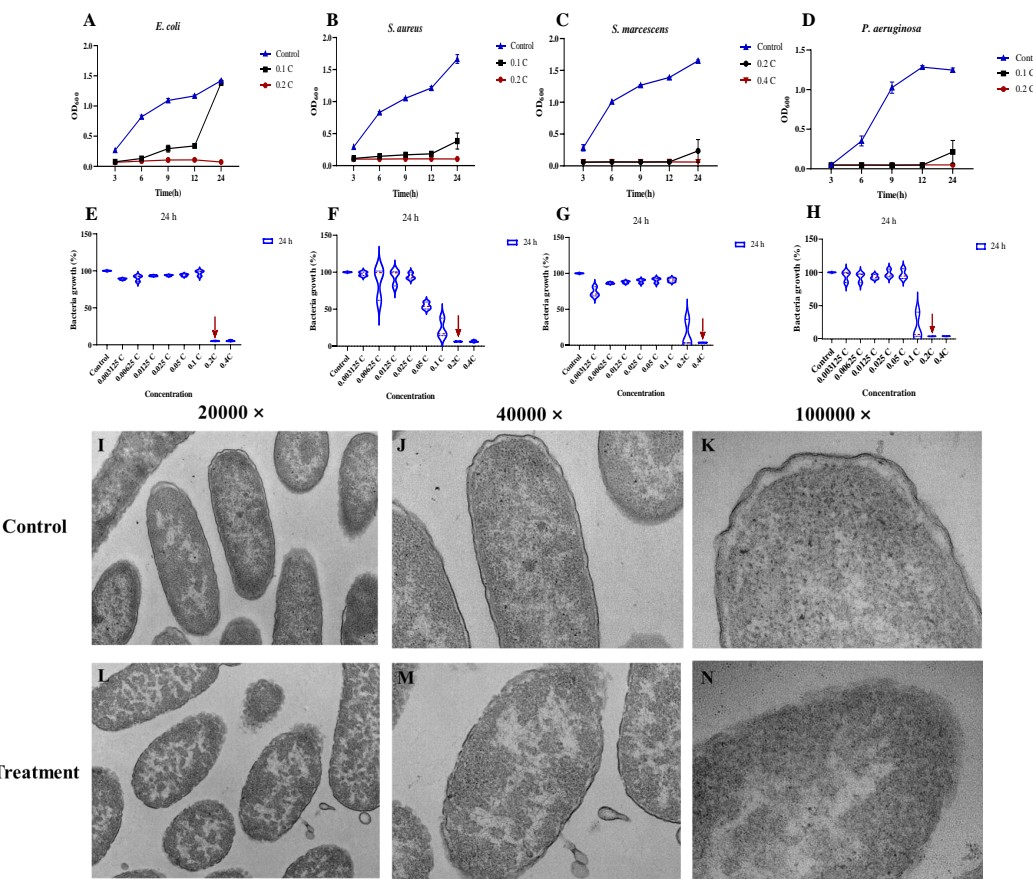

**Figure 1  Antibacterial activities of the MAC.** Antibacterial activities of the MAC. (A–D) Growth curves of *E. coli*, *S. aureus*, *S. marcescens* and *P. aeruginosa* at different times and the MAC concentrations, respectively. (E–H) MIC assays of different concentration gradients of the MAC against *E. coli*, *S. aureus*, *S. marcescens* and *P. aeruginosa* treated for 24 hours, respectively. The red arrow in each subfigure indicates the MIC that effectively inhibits visible growth of the corresponding microorganism. (I–N) TEM images of planktonic cells of *P. aeruginosa* treated with 0.2C MAC compared to the untreated controls.

(Figs. 1I–1J). Conversely, bacterial cells subjected to the MAC demonstrated significant cell wall and membrane impairments, accompanied by substantial intracellular material leakage (Figs. 1L–1N). These impairments caused alterations in the morphology of bacterial cells, which might lead to cell death.

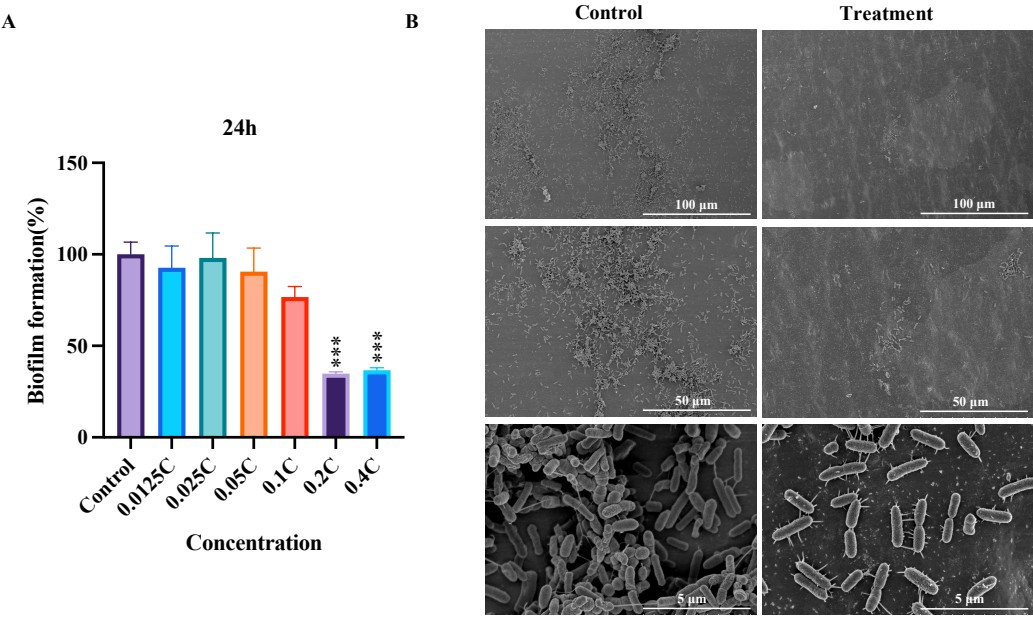

**Figure 2  The inhibition of biofilm by the MAC.** (A) The inhibition rate of biofilm formation in *P. aeruginosa* after 24 h of treatment with different concentrations of the MAC. (B) SEM images of biofilm formation after treatment with 0.2C MAC for 24 h (second vertical row) compared to the control group (first vertical row).

## The MAC can reduce biofilm formation

To evaluate the inhibitory potential of the MAC on biofilm formation in *P. aeruginosa*, we performed biofilm formation assays and SEM analysis. Our assays showed that all concentrations inhibited *P. aeruginosa* biofilm formation compared to the untreated control (Fig. 2A). The biofilm formation was significantly inhibited after the MAC treatment using 0.4C and 0.2C of the MAC. Our SEM analysis exposed a decrease in biofilm density after the 0.2C treatment (Fig. 2B). As shown in Fig. 2B, bacteria in control groups have wire-like structures, which potentially an extracellular DNA (eDNA) or matrix fibers (*Alhede et al., 2014*), that are the components of a biofilm matrix and useful for providing space for bacterial living and moving (*Panlilio & Rice, 2021*). In the treatment groups, the MAC drug cocktail can reduce the connections of those structures in *P. aeruginosa,* indicating that the MAC may be potential to directly act on the eDNA leading to denaturation and deactivation and affecting the synthesis of matrix fibers to reduce their connections. These findings also suggest that the MAC exhibits a considerable ability to effectively mitigate biofilm formation and potentially disrupt bacteria-bacteria interactions in *P. aeruginosa*.

## Whole-transcriptome sequencing and analysis

Illumina sequencing platform generated 103.3 million paired-end raw reads in three control group samples, and 102.9 million paired-end raw reads in three MAC-treated sample groups (Table S2). After preprocessing the raw reads (the removal of low-quality

reads), we had 14.9 GB (or 102.7 million of paired-end clean reads) and 15 GB (102.6 million paired-end clean reads) of data in the control and treatment groups, respectively (Table S3). The overall Q20 percentage was above 96%, the overall Q30 percentage was above 91%, and GC content was approximately 58% both in two sample groups. We also remapped all reads onto the reference genome sequence of *P. aeruginosa* ATCC 27853. At least 97.8% of total reads mapped to the reference genome for each sample from two groups with unique read mapping rate of at least 95% above (Table S4). These results indicate the high quality of the clean reads and are suitable for downstream analyses.

## Identification of differentially expressed genes (DEGs)

Before comparing transcriptomic profiles of the MAC treatment and the control groups. Principal component analysis (PCA) and sample normalization were performed for all samples (Fig. S2). PCA analysis result indicates a significant difference in gene expression levels between the two groups, providing evidence that the MAC treatment had an impact on the gene expression of *P. aeruginosa*. The normalized data indicates that the normalization of six samples worked well, and the data was suitable for comparison and differential expression analysis. By comparing the transcriptomic profiles of the two groups, we identified 1,093 differentially expressed genes (DEGs) consisting of 659 upregulated genes and 434 downregulated genes (Fig. 3A and Table S5). Among the 434 downregulated genes in the MAC treatment group, we found 62 genes displaying fold changes greater than four folds. Notably, a diverse range of enzyme-related activities was found among the highly downregulated genes, encompassing aldol/keto reductases (*ACG06_RS17845*, fold change (FC) = 9.25), putative hydro-lyases (*ACG06_RS15915*, FC = 7.96), ribose-phosphate pyrophosphokinase (*ACG06_RS12730*, FC = 7.35), betaine-aldehyde dehydrogenase (*ACG06_RS30685*, FC = 7.28), FMN-dependent NADH-azoreductase AzoR2 (*ACG06_RS16675*, FC = 7.27), uroporphyrinogen-III C-methyltransferase (*ACG06_RS02660*, FC = 6.38), tryptophan synthase subunit beta (*ACG06_RS00215*, FC = 6.37), MBL fold metallo-hydrolase (*ACG06_RS12740*, FC = 6.36), NADP-dependent glyceraldehyde-3-phosphate dehydrogenase (*ACG06_RS14485*, FC = 6.32), and various others. Moreover, an assortment of genes associated with transporters and translocators were also identified, such as LysE family translocators (*ACG06_RS10390*, FC = 7.13), multidrug resistance MFS transporters (*ACG06_RS04790*, FC = 10.68), and components of the type III secretion system (*ACG06_RS18070, 18090, 18005, 17955, 18045, 18035, 18055*; fold change are all greater than four times). Conversely, upregulated genes encompassed peroxiredoxins, known for their involvement in safeguarding cells against oxidative stress, along with several tRNA molecules crucial for protein synthesis (*ACG06_RS28405, 05490, 17535, 26955, 27425, 08615, 22155*; fold change are all greater than four times), namely tRNA-Leu, tRNA-Met, tRNA-Asp, and tRNA-Glu. Furthermore, the results revealed that the gene expression patterns among the samples following treatment with the MAC were consistent (Fig. 3B), indicating that the treatment might target specific biological processes or pathways within *P. aeruginosa*.
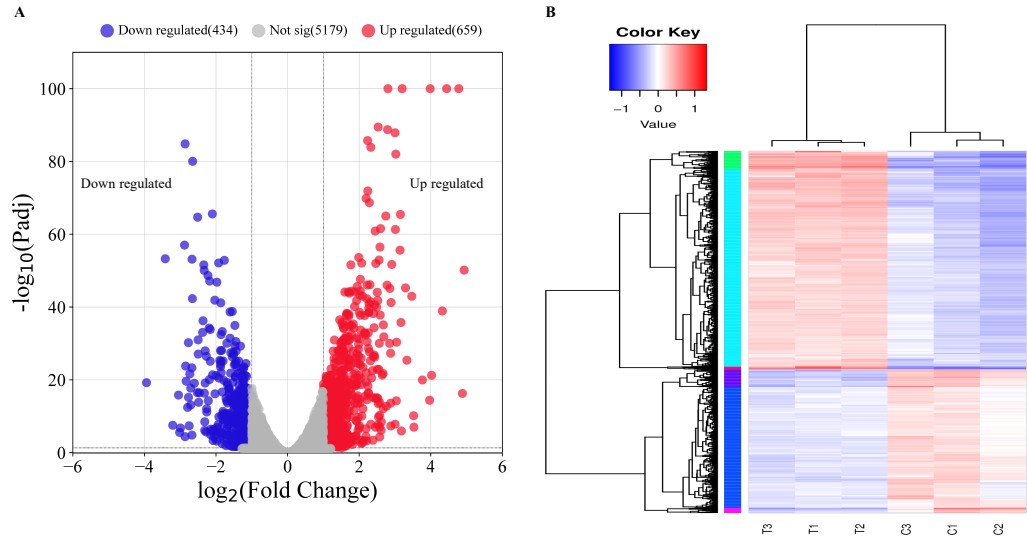

**Figure 3** **Overview of the differentially expressed gene analysis.** (A) Volcano plot for all DEGs. The horizontal axis shows the log2 (fold change) value, The vertical axis represents the mean expression value of log10 (Padj). Gray dots indicate no significant difference in genes, red dots indicate upregulated DEGs, and blue dots indicate downregulated DEGs. (B) Clustered heatmap of DEGs. Clustering was carried out with the value of log10(FPKM+1), with red indicating high-expression genes and blue indicating low-expression genes. The color from blue to red indicates higher gene expression.

## Gene ontology (GO) enrichment analysis

To gain insights into the functions of the DEGs to elucidate the antibacterial mechanisms of the MAC, we performed gene ontology (GO) term enrichment analyses separately on the sets of upregulated and downregulated differentially expressed genes (DEGs). The 30 most significantly enriched upregulated and downregulated GO terms were plotted (Figs. 4A, 4B). The upregulated genes enriched in the identified DEGs related to bacterial production capacity, specifically translation and ribosomes, suggesting increased protein synthesis activity. Additionally, the upregulation of genes involved in the catabolic processes of tyrosine and L-phenylalanine, tricarboxylic acid cycle, and oxidative phosphorylation indicated an increased energy production and metabolism, which likely contributed to the bacterial adaptation and survival strategies. On the contrary, the downregulated DEGs exhibited distinct functional patterns. Among the biological processes, the most significantly downregulated pathway was associated with biosynthetic and catabolic processes, including urea catabolic process, heme biosynthetic process, porphyrin-containing compound biosynthetic process, coenzyme biosynthetic process, and arginine catabolic process to succinate, with 58% of the genes assigned to these processes. Moreover, approximately 50% of the downregulated genes were associated with protein secretion, predominantly involving genes related to the type III secretion system, probably impacting bacterial pathogenesis and virulence. Regarding the molecular function, the most highly downregulated genes were mainly assigned to energy metabolism, such as cobalamin—transporting ATPase activity, GTPase activator activity, electron carrier

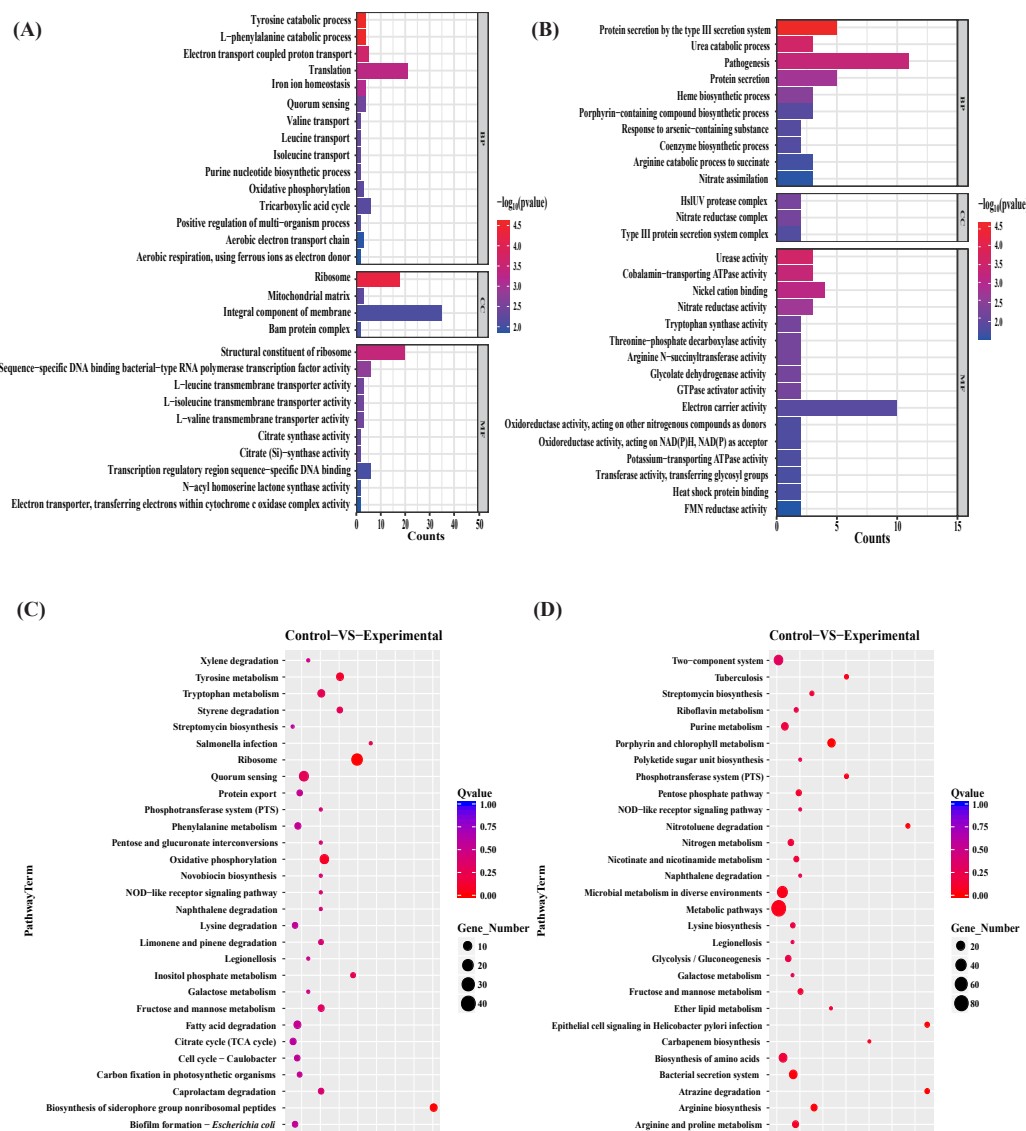

**Figure 4** **GO terms and KEGG pathway enrichment analyses.** The upregulated (A) and downregulated (B) difference gene of *P. aeruginosa* GO term. Biological pathways associated with the upregulated (C) and downregulated (D) genes after the MAC treatment against *P. aeruginosa*.

activity, oxidoreductase activity, Oxidoreductase activity, potassium—transporting ATPase activity, suggesting potential disruptions in these vital activities. Additionally, to better understand the enriched GO terms identified in the GO analysis, we conducted directed acyclic graph (DAG) analyses to generate DAG diagrams representing the molecular function, cellular component, and biological process (Figs. S3–S8).

## KEGG pathway enrichment analysis

To investigate the biological pathways regulated by the MAC, KEGG pathway enrichment analyses on the sets of upregulated and downregulated genes were performed (Figs. 4C, 4D). The exposure to the MAC had a compensatory effect on bacterial production capacity, likely aimed at preserving cellular homeostasis. Notably, a subset of genes associated with ribosome formation and aminoacyl-tRNA biosynthesis displayed significant upregulation (Fig. S9). Ribosomes are integral to protein synthesis, and their upregulation enables bacteria to generate an increased quantity of proteins. This augmented protein synthesis capability is pivotal for repairing damaged cellular components, synthesizing stress-related proteins, and sustaining essential cellular functions in demanding conditions (*Cheng-Guang & Gualerzi, 2021*). The intensified biosynthesis of aminoacyl-tRNAs, which ensures accurate attachment of amino acids to their corresponding tRNA molecules, aids bacteria in maintaining fidelity in protein synthesis and adapting to the fluctuating environment (*Laursen et al., 2005*). On the contrary, our results suggested that the MAC inhibited the growth of *P. aeruginosa* by suppressing various bacterial activities involved in metabolism (metabolic pathways, riboflavin metabolism, purine metabolism, pentose phosphate pathway, nitrogen metabolism, nicotinate and nicotinamide metabolism, microbial metabolism in diverse environments, galactose metabolism, fructose and mannose metabolism, ether lipid metabolism, arginine and proline metabolism, and arginine and proline metabolism), biosynthesis (streptomycin biosynthesis, polyketide sugar unit biosynthesis, lysine biosynthesis, glycolysis/gluconeogenesis, carbapenem biosynthesis, and arginine biosynthesis), biodegradation (nitrotoluene degradation, naphthalene degradation, and atrazine degradation), and transportation (bacterial secretion system and phosphotransferase system). Notably, the MAC downregulated the expression of almost all genes associated with the Type III secretion system, porphyrin metabolism, and two-component system (Fig. S10). These findings highlight the comprehensive inhibitory effects of the MAC on various cellular processes that are essential for bacterial growth, survival, and pathogenesis.

## Validation of DEGs by real-time RT-PCR

To validate the list of DEGs identified in our RNA-Seq analysis, RT-PCR was utilized to assess the expression levels of nine randomly selected DEGs. Our RT-PCR results showed that the expression level of *ahpC, feoA, gatC, cobU*, and *pfeR* were upregulated, while the expression levels of *pscF, pscP, glcF, cobA*, and *azoR2* were downregulated. The qRT-PCR and RNA-Seq data exhibit a high degree of concordance ($R^2 = 0.82$), suggesting that they are correlated and providing support to the list of DEGs identified in our RNA-Seq analysis (Fig. 5).

## DISCUSSION

Here we investigated that the MAC exhibited antibacterial activity against various bacteria including *E. coli*, *S. aureus*, *P. aeruginosa*, and *S. marcescens*, as supported by evidence from the MIC assay, MBC assay and TEM study. Moreover, the MAC exhibited biofilm inhibition activity, indicating that they have potential as therapeutic agents against persistent infections

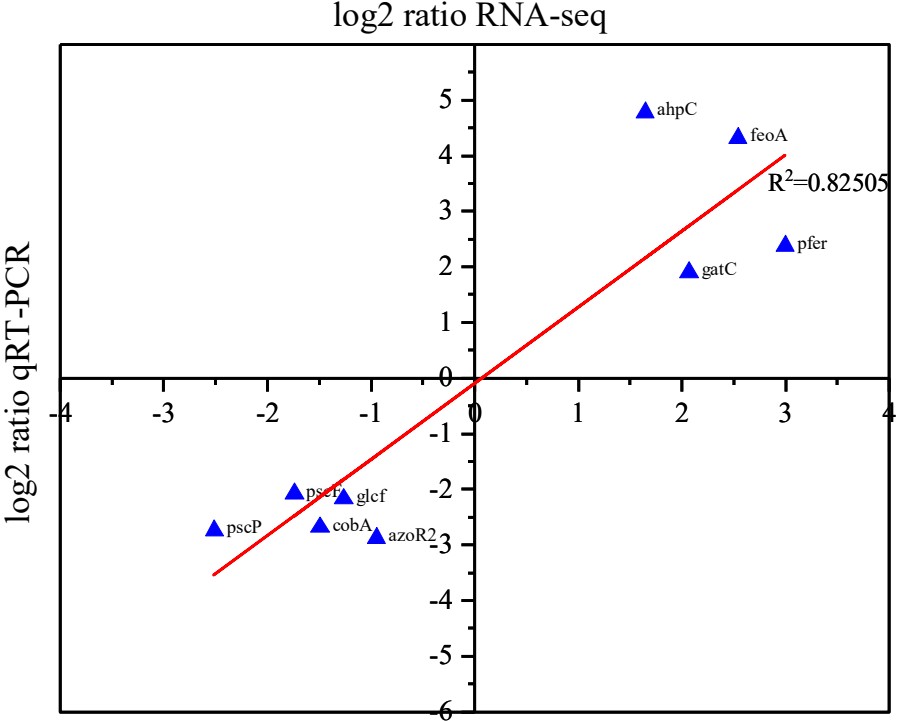

**Figure 5** **Correlation between RNA-Seq and RT-PCR data.** The $X$-axis of the plot represents the $\log_2$ (fold change) of RNA-Seq data, while the $Y$-axis represents the $\log_2$ (fold change) of RT-PCR data. The $R^2$ value refers to the coefficient of determination of the regression line.

and biofilm-causing human diseases. To further explore the antibacterial mechanism of the MAC, we conducted RNA-Seq analysis on *P. aeruginosa*. The results suggested that the antimicrobial action of the MAC is likely due to the suppression of key bacterial processes involved in metabolism, transport, signal transduction, and stress response which could reduce the survival of bacteria.

The metabolic activity of bacteria is a crucial indicator of their growth and viability (*Stokes et al., 2019*), making the expression of genes involved in metabolism an essential metric for assessing the viability of *P. aeruginosa*. In our study, several metabolic pathways were found to be downregulated, among which arginine biosynthesis and porphyrin metabolism were the most affected. Arginine is an important metabolite for maintaining bacterial growth, and it participates in metabolic pathways related to biofilm formation, toxicity, and antibiotic resistance (*Scribani Rossi et al., 2022*). Cobalamin, or vitamin B12, is a porphyrin derivative that serves as a cofactor for enzymes involved in a range of metabolic pathways, including methionine synthesis, fatty acid metabolism, DNA synthesis, and cell division (*Raux, Schubert & Warren\*, 2000*). The antibacterial mechanism of organic acids has been demonstrated to be associated with pH reduction, anionic toxicity, osmotic pressure, and disruption of the transmembrane proton gradient (*Carpenter & Broadbent, 2009*; *Cherrington et al., 1991*). Lower pH levels can inhibit enzyme activity, resulting in the downregulation of metabolic pathways. Thus, we speculated that the MAC affects the

growth and viability of *P. aeruginosa* by inhibiting the expression of metabolism-related genes, possibly attributed to the altered pH levels caused by the MAC which are mainly organic acids.

Interestingly, the MAC may attenuate the pathogenicity of *P. aeruginosa* towards organisms. The Type III secretion system (T3SS) is a specialized protein secretion system used by gram-negative bacteria to deliver major virulence factors (*Hauser, 2009*). The T3SS is known to be essential for *P. aeruginosa* as it allows them to interfere with the host cell signal transduction and evade host defenses (*Vance, Rietsch & Mekalanos, 2005*). Our results showed that the MAC down-regulated key genes, including genes *pscC, E, F, G, L, N, O, P, Q, R, T, U, X*, and *dnaK*, that encode proteins responsible for the formation and stability of the T3SS needle structure (*Galle, Carpentier & Beyaert, 2012*). Several genes encoding regulatory proteins (*PopN, PcrD, PpkA*) and chaperone protein (*PscT*) were also inhibited. Moreover, the gene *UreC*, which encodes the urease enzyme and is essential for *P. aeruginosa* to colonize acidic environments by neutralizing acidic conditions, was also downregulated (*Collins & D'Orazio, 1993*). Collectively, it is possible that the combined inhibition of multiple genes may have a cumulative impact on reducing the toxicity and fitness of *P. aeruginosa* and its ability to initiate and establish infections.

Our findings indicated that the MAC treatment resulted in the downregulation of genes associated with two-component systems (TCSs), which are known to play a crucial role in the process of biofilm formation in *P. aeruginosa*. For instance, TCSs are involved in regulating the transition of bacterial behavior from motile to stationary during biofilm formation, primarily through the production of extracellular structures, including appendages such as flagellum and fimbriae, as well as polysaccharides (*Mikkelsen, Sivaneson & Filloux, 2011*). Several genes *pilK, pilJ, motY*, ACG06_RS15830, RS15835, *FliA, siaC* related to the regulation of the biogenesis of the fimbriae, flagellin synthesis, extracellular polymeric substances synthesis, and twitching motility, were inhibited (*Mattick, Whitchurch & Alm, 1996*; *Ryder, Byrd & Wozniak, 2007*). These genes are involved in the formation of extracellular appendages like pili and flagella. The reduction of wire-like structures (eDNA or matrix fibers) between *P. aeruginosa* observed in the SEM experiment (Fig. 2B) following MAC treatment may be associated with the downregulation of genes related to TCSs involved in biofilm formation. We speculate that MAC treatment may inhibit tyrosine kinases and cyclic di-GMP enzymes in TCS signaling, as well as proteases and nucleases involved in the regulation of these appendage-related genes (*Wongkaewkhiaw et al., 2020*). This downregulation may disrupt or imbalance the TCS signaling pathway, thereby affecting the generation and connectivity of wire-like structures during biofilm formation, which is dependent on extracellular appendages (*Kaushik et al., 2022*). Specifically, MAC treatment may reduce the connectivity of wire-like structures by inhibiting the expression of genes involved in fimbriae biogenesis, flagellin synthesis, extracellular polymeric substances synthesis, and twitching motility. Our results suggest that the MAC treatment can influence the TCSs-mediated shift from motility to stationary behavior by altering the expression of genes related to extracellular appendages, which are crucial for biofilm formation in *P. aeruginosa*.
Bacteria possess the capability to enhance their resistance mechanisms and survive under extreme environmental conditions. The upregulation of resistance genes in response to stress enables bacteria to adapt and withstand the challenges posed by factors such as temperature, pH changes, and toxic substances (*Šeputienė et al., 2003*). The observed recovery of growth in the sublethal concentrations of the MAC (0.1 C) *E. coli* and *S. marcescens* back to control levels at 24 h suggests the possibility of adaptive responses. Such adaptations may occur gradually and enable the restoration of bacterial growth, ultimately resulting in a convergence of growth levels between the MAC-treated group and the controls (*Marquis et al., 1987*). Our KEGG pathway analysis of *P. aeruginosa* revealed that under the MAC stimulation, the aminoacyl-tRNA biosynthesis and ribosomal pathways were significantly upregulated, indicating that *P. aeruginosa* engages in positive feedback regulation to compensate for the lack of metabolic activities. The upregulated genes such as *ACG06_RS28405*, *RS08615*, *RS28410*, RS03440, RS26955, and *RS09260* encode corresponding tRNA synthetase that catalyzes the attachment of amino acids to tRNA molecules (*Ibba & Söll, 2000*). These findings suggest that bacteria actively work to ensure accurate and efficient protein translation, despite the environmental stress induced by drug exposure. The genes encoding the 30S ribosomal subunit proteins include *rpsD, F, H, J, K, L, M, T, U*, and *ykgO*, which are involved in translation initiation and elongation, as well as stabilizing the structure of the ribosome (*Nomura, Morgan & Jaskunas, 1977*). The genes encoding the 50S ribosomal subunit proteins include *rpmE, J* and *rplC, E, J, K, M*, which are important for ribosome assembly, peptide bond formation, and translation elongation (*Nomura, Morgan & Jaskunas, 1977*). Furthermore, our GO functional enrichment analysis indicated an enrichment of translation-related genes, indicating that bacterial cells overproduced various proteins such as enzymes, membrane proteins, and transporters to maintain protein synthesis and compensate for metabolic activity deficiency.

The results obtained from our experimental investigations offer valuable insights into the potential antibacterial effects of the MAC. The comparable antibacterial activity on *P. aeruginosa* observed between the MAC drug cocktail and ciprofloxacin is noteworthy. It indicates that the MAC cocktail, composed of malic acid, citric acid, glycine, and hippuric acid, possesses potent antimicrobial properties against *P. aeruginosa*. These results support the notion that the MAC cocktail could be a viable alternative or supplementary option to conventional antibiotics for treating *P. aeruginosa* infections. Notably, using the MAC cocktail as a treatment option may offer advantages such as cost-effectiveness, accessibility, and reduced likelihood of inducing antibiotic resistance. However, there is still scope for further improvement and exploration in this field. Firstly, by expanding the study to include a broader range of strains, especially those from clinical settings and ESKAPE pathogens, we can evaluate the applicability and effectiveness of MAC treatment in contexts that are more clinically relevant. This broader study scope would provide invaluable insights into the potential clinical implications of MAC treatment and its impact on different bacterial strains. Secondly, it is crucial to evaluate the *in vivo* antibacterial and antibiofilm efficacy of the MAC using different animal infection models particularly mimicking lung fibrosis. Such studies would provide a more comprehensive understanding of the MAC's effectiveness in

real-life scenarios. Thirdly, given the intricate nature of microorganisms, it is imperative to conduct further comprehensive research to evaluate the applicability of the MAC against a diverse range of bacterial species, as well as other nonbacterial microorganisms such as fungi and viruses (*Coban, 2020*; *Li et al., 2002*; *Shokri, 2011*). Finally, it is also imperative to consider the intricate dynamics of infections *in vivo* when assessing the translational potential of this research (*Tängdén et al., 2020*). This investigation would enable us to assess the efficacy of the MAC in realistic infection scenarios and evaluate any potential risks posed to the host. Such endeavors will undoubtedly contribute to the advancement of antimicrobial strategies and the reduction of risks associated with infections.

## CONCLUSION

We have demonstrated the potent bacteriostatic activity of the MAC drug cocktail against several bacteria species, particularly *P. aeruginosa*, comparable to the positive control ciprofloxacin. The inhibitory effect of the MAC on the formation of *P. aeruginosa* biofilms was also established. Our RNA-Seq analysis revealed that the MAC likely targets key bacterial functions, such as metabolism, secretion system, signal transduction, and cell membrane. Additionally, the pathogenicity of *P. aeruginosa* was likely impaired by the presence of the MAC. This study provides important insights into the antibacterial activity, antibiofilm activity and mechanism of the MAC drug cocktail, which is a potential alternative for antibiotics to treat human infectious diseases in the future.

## ACKNOWLEDGEMENTS

We thank the Laboratory and Research Center of Wenzhou-Kean University for providing facility support, including space and equipment. We greatly thank the Office of Research and Sponsored Program (ORSP) of Wenzhou-Kean University for their project management. We also appreciate Dr. Aloysius Wong and Dr. Bo Zhang from Wenzhou-Kean University providing bacterial strains used in this study and guidance on bacterial tests.

### Funding

This work was funded by the high-level talent recruitment program for academic and research platform construction (Reference Number: 5000105) from Wenzhou-Kean University. This study was also supported by the Wenzhou Municipal Key Lab for Biomedical and Biopharmaceutical Informatics of Wenzhou-Kean University (20211227000125). This study was also supported by the Wenzhou Science and Technology "Qing Miao Project". The funders had no role in study design, data collection and analysis, decision to publish, or preparation of the manuscript.

### Grant Disclosures

The following grant information was disclosed by the authors:

The high-level talent recruitment program for academic and research platform construction (Reference Number: 5000105) from Wenzhou-Kean University.

The Wenzhou Municipal Key Lab for Biomedical and Biopharmaceutical Informatics of Wenzhou-Kean University: 20211227000125.

The Wenzhou Science and Technology ''Qing Miao Project''.

## Competing Interests

The patents mentioned in this study were invented by the authors, and the authors declare no conflict of interest.

## Author Contributions

- Kunping Song performed the experiments, analyzed the data, prepared figures and/or tables, authored or reviewed drafts of the article, and approved the final draft.
- Li Chen performed the experiments, analyzed the data, prepared figures and/or tables, authored or reviewed drafts of the article, and approved the final draft.
- Nanhua Suo performed the experiments, analyzed the data, prepared figures and/or tables, authored or reviewed drafts of the article, and approved the final draft.
- Xinyi Kong performed the experiments, analyzed the data, prepared figures and/or tables, authored or reviewed drafts of the article, and approved the final draft.
- Juexi Li performed the experiments, analyzed the data, prepared figures and/or tables, authored or reviewed drafts of the article, and approved the final draft.
- Tianyu Wang performed the experiments, analyzed the data, prepared figures and/or tables, authored or reviewed drafts of the article, and approved the final draft.
- Lanni Song performed the experiments, analyzed the data, prepared figures and/or tables, and approved the final draft.
- Mengwei Cheng performed the experiments, analyzed the data, prepared figures and/or tables, and approved the final draft.
- Xindian Guo performed the experiments, analyzed the data, prepared figures and/or tables, and approved the final draft.
- Zhenghe Huang performed the experiments, analyzed the data, prepared figures and/or tables, and approved the final draft.
- Zichen Huang performed the experiments, analyzed the data, prepared figures and/or tables, and approved the final draft.
- Yixin Yang analyzed the data, authored or reviewed drafts of the article, and approved the final draft.
- Xuechen Tian conceived and designed the experiments, analyzed the data, prepared figures and/or tables, authored or reviewed drafts of the article, and approved the final draft.
- Siew Woh Choo conceived and designed the experiments, analyzed the data, authored or reviewed drafts of the article, and approved the final draft.

## Patent Disclosures

The following patent dependencies were disclosed by the authors:

China Invention Patent Number CN202111195294.4 and the Luxembourg Invention Patent Number LU102887.

## Data Availability

The raw sequence data is available at NCBI: PRJNA977404.

Additional raw data is available at Figshare: Tian, Xuechen (2023). Raw data for the publication of Whole-transcriptome analysis reveals mechanisms underlying antibacterial activity and biofilm inhibition by a malic acid combination (MAC) in *Pseudomonas aeruginosa*. figshare. Dataset. https://doi.org/10.6084/m9.figshare.23524185.v2

## Supplemental Information

Supplemental information for this article can be found online at http://dx.doi.org/10.7717/peerj.16476#supplemental-information.

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
