# Peer review of "Whole-transcriptome analysis reveals mechanisms underlying antibacterial activity and biofilm inhibition by a malic acid combination (MAC) in Pseudomonas aeruginosa"

_PeerJ, doi:10.7717/peerj.16476_

## Round 0.1 · original submission · Major Revisions

When preparing a revised version of the manuscript, please consider all the comments made by the referees. I agree with the reviewers that further experiments are necessary, explicitly using clinically significant bacterial strains. It is also essential to include antibiotics as positive controls for antibacterial activity and report on the MIC (minimal inhibitory concentration) and MBC (minimal bactericidal concentration). Testing the proposed MAC antibacterial product on ESKAPE pathogens would be ideal to demonstrate its relevance.

·

Basic reporting

The manuscript clearly states its objectives, and the results are significant. The theme is appropriate for PeerJ scope. However, there are details that can be improved in the manuscript:
1.- The present results are important and provide insights into the possible mechanism involved, from a transcriptomic perspective. However, it is crucial to correctly describe the results. For example, in the table 5S and Fig. 3A, the downregulated gen does not reach the four fold change (Lines 303-304), and it is recommended to include the fold values for the expression of each gene (Line 306-311).
2.- The UreB gene (Line 415) is not listed in the upregulated genes table (Table 5S). Moreover, the genes described in Lines 425-428, only pilJ gene is identified in table 5S.
3.- (Lines 404-406) Can any organic acid that induces pH reduction promote the same effect observed during MAC treatment? Was the reduction of pH evaluated in a system similar to the experimental one used?
4.- Line 363.- Do Pseudomonas produce chlorophyll?
5.- (Fig. 1) It is recommended to incorporate the growth curve of S. marcescens at 0.4C.
6.- (Fig. 2A).- The axis title “Biofilm inhibition (%)” is confusing. Consider using "biofilm formation (%)" for more clarity.
7.- (Fig. 2A) Does MAC at 0.05 and 0.1C promote biofilm formation? What is the fundament behind this MAC effect? It is recommended to describe and discuss this MAC effect.
8.- (Fig. 2B).- The scale reference is not visible. Suggest increasing its size. What are the scale differences between left and right images?
9.- Line 267.- “P. aeruginosa” Itallic.

Experimental design

1.- The experimental design is adequate for the objectives. However, the section titled “Assessing bacterial ultrastructural alterations through transmission electron microscopy” is unclear and confusing. It is necessary to provide a reference for this methodology.
2.- (Line 208).- What is using the rRNA removal kit?
3.- (Line 165).- Better describe the “uranyl acetate 50% ethanol saturated” solution.
4.- (Line 204).- Trizol.
5.- (Line 251).- It is advisable to define or differentiate the SEM abbreviation since it has been used previously.
6.- Line 111, 143, 173.- mL
7.- Line 125.- The term “revived” is not convenient.

Validity of the findings

The discussion of the results reveals significant aspects of gene expression changes, but it is necessary to discuss these results in relation to the MAC composition and correlate them with other in vitro research that studies biofilm inhibition. Furthermore, it is important to explore the possibility of biofilm inhibition through enzyme inhibition, which is indispensable for biofilm formation. By incorporating these aspects into the discussion, the overall understanding and applications of the findings can be enhanced.

Reviewer 2 ·

Basic reporting

no comment

Experimental design

Many natural substances have antimicrobial activity, including organic acids. The authors present the effect of the MAC mixture consisting of 1.2% malic. acid, 0.2% citric acid, 0.1% glycine, and 0.3% hippuric acid.

Unfortunately, the publication needs corrections:
1. The authors used only 4 strains of bacteria and these are reference strains. Clinical strains for which MICs may differ from reference strains should also be tested.
2. In the methodology, the authors wrote about "Mackay's turbidity standard", which is probably a mistake and should be "McFarland turbidity standard".
3. The biggest problem is the lack of control, i.e. substances with proven antibacterial properties, e.g. an antibiotic. MIC and MBC should be reported in ug/mL or mg/L. The use of an antibiotic as a control will make it possible to compare the results and determine whether the tested MAC mixture has a real chance of being used in the fight against pathogens. For example, the value 0.2C in Table 1 currently represents nothing. If the value is given in ug/mL and there is a result for eg cefotaxime and gentamicin next to it, then you can see if the MAC mixture has a super effect or if the effect is medically insignificant.
4. In Figure 1 and 2 there is no description in the legend what the red arrow means.
5. Figures have poor quality.
6. Conclusions should be corrected according to new results with antibiotics as controls.

Validity of the findings

no comment

Reviewer 3 ·

Basic reporting

xxxx

Experimental design

xxxx

Validity of the findings

xxxxx

Additional comments

1. The authors refer to a screening carried out previously, but these data are not published (referred to as ‘Tian et al, 2023, unpublished data’). These data are however essential to understand the data reported here.

2. Why is the combination of malic acid, citric acid, glycine and hippuric acid called a ‘MAC’? What is the rationale behind the concentrations used? What is the effect of the individual compounds?

3. What is the pH of the MAC and the medium after addition of MAC? Did the authors rule out that differences in pH contribute to the effects being observed?

4. The biofilm experiments are very basic (conventional medium, 96-well plate, crystal violet staining) and it is not very clear what the data presented in Figure 2A mean. Also, I would assume that it is straight forward that biofilm formation is inhibited when compounds are added at a concentration that is at or above the MIC.

5. While relevant organisms are studied, the work is limited to reference strains only – inclusion of (recent) clinical isolates would be useful. Bacteria were grown under conditions that do not mimick physiological conditions at all (TSB, plastic microtiter plate).

---

## Round 0.2 · Minor Revisions

Please be encouraged by the prompt reviews, and we look forward to your revised manuscript.

·

Basic reporting

The authors provided clear responses to the previously mentioned comments, addressing each one comprehensively. However, they have yet to address the following comment:

7.- (Fig. 2A) Does MAC at 0.05 and 0.1C promote biofilm formation? What is the fundament behind this MAC effect? It is recommended to describe and discuss this MAC effect.

I consider the complete description of this result (Fig. 2A) and its corresponding discussion to be indispensable for understanding the antibiofilm effect conferred by MAC. This result highlights that applying MAC treatment at low concentrations promotes biofilm formation. Does this hold any clinical significance?

Experimental design

The changes applied to the Experimental design are appropriate.

Validity of the findings

The authors speculate that MAC treatment may inhibit tyrosine kinases among other potential aspects. However, it is advisable to incorporate relevant citations from the bibliography to support this discussion.

Reviewer 2 ·

Basic reporting

The authors corrected the manuscript according to my suggestions. I recommend the article for publication.

Experimental design

The authors corrected the manuscript according to my suggestions. I recommend the article for publication.

Validity of the findings

The authors corrected the manuscript according to my suggestions. I recommend the article for publication.

---

## Round 0.3 · accepted · Accept

Thanks for addressing the revisions requested. Now, your manuscript is accepted in PeerJ.